# Spinal Solitary Fibrous Tumors: An Original Multicenter Series and Systematic Review of Presentation, Management, and Prognosis

**DOI:** 10.3390/cancers14122839

**Published:** 2022-06-08

**Authors:** Caroline Apra, Amira El Arbi, Anne-Sophie Montero, Fabrice Parker, Steven Knafo

**Affiliations:** 1Sorbonne Université, 75013 Paris, France; anne-sophie.montero@aphp.fr; 2Neurosurgery Department, Pitie Salpêtrière Hospital, 75013 Paris, France; 3Neurosurgery Department, Bicêtre Hospital, 94270 Kremlin-Bicêtre, France; mira.elarbi@gmail.com (A.E.A.); fabrice.parker@aphp.fr (F.P.); 4University Paris-Saclay, 91190 Gif-sur-Yvette, France

**Keywords:** spine, medulla, intramedullary, solitary fibrous tumor, hemangiopericytoma, neurosurgery, STAT6

## Abstract

**Simple Summary:**

Solitary fibrous tumors are rare benign or cancerous tumors that develop in all tissues, including close to the spinal cord. These cases are exceptional and we describe their presentation and outcome based on 31 published cases and 10 patients on whom we operated. The tumors can develop in any portion of the spine and cause back pain, associated with neurological deficits, such as compression of a nerve or the spinal cord, in 66% of patients. Surgical removal is the first step towards diagnosis and treatment, but complete removal could be achieved in only 70% of patients, due to bleeding or spinal cord invasion. Tumors were found to recur after a mean 5.8 years (1 to 25), without identified risk factors. However, in patients with subtotal removal, radiotherapy significantly improves the rate of recurrence. In total, spinal solitary fibrous tumors are treated by neurosurgeons on the front line but discussion in a multidisciplinary team will provide general treatments, especially radiotherapy after subtotal removal.

**Abstract:**

All solitary fibrous tumors (SFT), now histologically diagnosed by a positive nuclear STAT6 immunostaining, represent less than 2% of soft tissue sarcomas, with spinal SFT constituting a maximum of 2% of them, making these tumors extremely rare. We provide an up-to-date overview of their diagnosis, treatment, and prognosis. We included 10 primary STAT6-positive SFT from our retrospective cohort and 31 from a systematic review. Spinal pain was the most common symptom, in 69% of patients, and the only one in 34%, followed by spinal cord compression in 41%, radicular compression, including pain or deficit, in 36%, and urinary dysfunction specifically in 18%. Preoperative diagnosis was never obtained. Gross total resection was achieved in 71%, in the absence of spinal cord invasion or excessive bleeding. Histologically, they were 35% grade I, 25% grade II, and 40% grade III. Recurrence was observed in 43% after a mean 5.8 years (1 to 25). No significant risk factor was identified, but adjuvant radiotherapy improved the recurrence-free survival after subtotal resection. In conclusion, spinal SFT must be treated by neurosurgeons as part of a multidisciplinary team. Owing to their close relationship with the spinal cord, radiotherapy should be considered when gross total resection cannot be achieved, to lower the risk of recurrence.

## 1. Introduction

In 2016, the World Health Organization (WHO) introduced the combined term “solitary fibrous tumor/hemangiopericytoma” for describing connective tissue tumors of the central nervous system with positive STAT6 nuclear immunostaining, which was replaced by “solitary fibrous tumors” (SFT) alone in 2021, to conform fully with soft tissue pathology nomenclature [1]. The grouping of these two entities, which were separated until then, is grounded in an overlapping histological description, associated with a shared genetic signature: the fusion of the NGFI-A-binding protein 2 (*NAB2*) and signal transducer and activator of transcription 6 (*STAT6*) genes due to an inversion at chromosome 12q13, which is a hallmark for all SFT, regardless of their localization, since its first description in 2013 [2]. SFT represent less than 2% of soft tissue sarcomas, with central nervous system SFT constituting 20% of SFT, with only one spinal SFT for every ten intracranial lesions, making these tumors extremely rare in clinical practice [2]. We performed a retrospective multicenter series of spinal SFT, focusing on STAT6-positive tumors only, and added a systematic review of all published cases. The aim of this review is to provide an up-to-date overview of the diagnosis, treatment, and prognosis of these rare tumors, with a discussion about individual clinical decision making.

## 2. Materials and Methods

### 2.1. Original Series

All patients with a histological diagnosis of spinal SFT who underwent surgery in the neurosurgical departments of the Bicêtre and Pitié-Salpêtrière hospitals in Paris, France, between 1988 and 2020 were included. SFT diagnosis was confirmed by two expert neuropathologists with confirmed positive STAT6 nuclear immunostaining even for the cases dated before 2016. Medical records were reviewed for clinical and radiological presentation, histopathologic features, surgical treatment, postoperative therapies, and outcomes. Tumors were graded radiologically [3]: I, extradural type; II, intradural type; and III, intra- to extradural and paravertebral type. Extent of resection was estimated from the operative reports and postoperative MRI. Tumors were graded histologically according to the 2016 WHO classification [1]. Duration of follow-up was calculated as the duration from the date of surgery to the last outpatient department visit. Recurrence was defined as local tumor growth on MRI, whether symptomatic or not. Ethical approval was granted by the French Neurosurgical Society review board (IRB00011687; 2022/14).

### 2.2. Systematic Review

The literature review was performed according to the PRISMA checklist [4]. The database (PubMed) was searched for the combination of terms “spine”, “spinal”, “medulla”, “medullary”, “solitary fibrous tumor”, “hemangiopericytoma” in December 2021. Eligible articles reported at least one case of spinal SFT. Exclusion criteria were the absence of STAT6-positive nuclear immunostaining, spinal metastatic localizations, and the absence of clinical data. Clinical, radiological, and histological data, including perioperative descriptions of the tumors, were collected.

### 2.3. Statistical Analysis

All statistical analyses were performed using Microsoft Excel (version 2202). For analyzing risk factors for recurrence, exact Fisher test was performed due to the small size of the population. Schematic figures were created with http://www.BioRender.com (accessed on 10 April 2022).

## 3. Results

### 3.1. Patients

We included 10 cases from our own retrospective cohort and retrieved 31 cases from the systematic review, as detailed in the Appendix A (Appendix A) [5,6,7,8,9,10,11,12,13,14,15,16,17,18,19,20]. All were primary SFT with a STAT6-positive pathological diagnosis. Most information was available from all cases, as detailed in Appendix A. Results are given as: all patients [review; cohort]. There were slightly more women (51%, 55%/40%, sex ratio 1.05) and the mean age at diagnosis was 46 (45/47), ranging from 10 to 81 (10–81/32–71). All results are detailed in Table 1.

### 3.2. Clinical Presentation

Spinal pain was the most common symptom, found in 27 patients (69% [76%; 50%]), and as an isolated symptom in 14 (34% [45%; 0%]). Radicular symptoms, including pain or deficit, were present in 14 patients (36% [28%; 60%]), whereas spinal cord compression, including increased reflexes or sensory–motor deficit, was present in 16 patients (41% [31%; 70%]), and urinary dysfunction was reported in seven patients (18%, [10%; 40%]). Symptoms typically worsened gradually, with a mean duration of clinical symptoms before surgery of 18 months (20/10), ranging from less than 1 month to 11 years. For patients with isolated spine pain, the mean duration of symptoms before surgery was 15 months.

### 3.3. Initial Radiological Findings

All patients underwent a preoperative spine MRI with contrast. Although T1 and T2 MRI aspects varied, all tumors showed marked homogeneous or heterogeneous enhancement by gadolinium. No calcification or acute intratumoral hemorrhage was observed. CT scan was not systematically available on retrieval, although it was likely performed in clinical practice before surgery. In a few cases, scalloping was observed in extracanalar tumors, but no exostosis. In our cohort, three patients had a preoperative angiography without embolization. Tumors were cervical in 12 patients (29% [23%; 50%]), thoracic in 21 patients (51% [54%; 40%]), and lumbar in eight patients (20% [23%; 10%]), which is proportional to the length of each segment, cervical spine vertebras constituting 30%, thoracic 50%, and lumbo-sacral 20% of the whole spine. Lesions involved one to two vertebrae in most patients (93% [97%; 80%]). Spinal SFT were type I in two patients (9% [14%; 0%]), II in 15 patients (65% [57%; 78%]), and III in six patients (26% [29%; 22%]). At least four [2; 2] extracanalar lesions showed extension in the foramina, giving them a dumbbell aspect.

### 3.4. Operative Findings

All patients underwent first-line surgery, through an isolated posterior approach with laminectomy, or associated with an anterior approach, an arthrodesis or a thoracoscopy in one case, depending on the tumor extension. Gross total resection was achieved in 29 patients (71% [71%; 70%]). The reasons for subtotal resection included spinal cord invasion and excessive bleeding. The use of neuro-monitoring was mentioned only once, after recurrence. Operatively, tumors were identified as being purely extramedullary in patients (63% [68%; 50%]) with or without description of dural and pial invasion, whereas other cases invaded the medulla. No intraoperative complication other than bleeding was noted, either in the literature or in our series. A video showing perioperative observations is available as a Appendix A.

### 3.5. Histological Findings

All tumors were diagnosed as spinal SFT with positive STAT6 nuclear immunostaining. Seven [6; 1] were classified as grade I, five [0; 5] as grade II, and eight [5; 3] grade III. The mitoses count ranged from 0 to 15 per 10 high-power fields, and Ki67 from 0% to 15%. Other reported immunostainings include CD34, vimentin, proteinS100, EMA, and SMA, but these are not reliable in SFT diagnosis. No histological evidence of medullary invasion was described.

### 3.6. Adjuvant Treatment and Outcome

No complication, either infection, cerebrospinal fluid leakage, neurological worsening, or death, was reported after surgery. Eleven (27 [23%; 40%]) patients received immediate postoperative adjuvant treatments, including 11 radiotherapy and one neoadjuvant chemotherapy. The rationale for performing these treatments was not systematic and not explicit but these patients included one case of grade III SFT and four cases of subtotal resection. Available follow-up ranged from 12 months to 30 years for 35 [26; 9] patients. Recurrence was observed in 15 patients (43% [42%; 44%]), after a mean 5.8 years, ranging from 1 to 25 years, as detailed in Figure 1. No significant risk factor for recurrence could be identified, but there was a tendency to recur for patients with incomplete surgery or no adjuvant radiotherapy (Table 2). Survival analyses show that adjuvant radiotherapy significantly improves the recurrence-free survival in patients with subtotal resection but has no effect in patients with gross total resection (Figure 1). Repeated recurrences were observed in some cases, but data were scarce. After recurrence, nine (75% [63%; 100%]) patients underwent a second surgery, with a combination of radiotherapy, carbon-ion radiotherapy, or proton therapy, and one was treated with Pazopanib for progressive disease after third recurrence. There were no systematic data about metastasizing.

## 4. Discussion

### 4.1. Clinical Approach to Spinal SFT Management Based on Data and Experience

Spinal SFT are extremely rare tumors, and patients usually present with nonspecific symptoms. Most neurosurgeons will probably operate zero to a few spinal SFT in their career, without even suspecting it before histological results. The clinical and radiological features of spinal SFT are summarized in Figure 2. The first step toward diagnosis is to perform a spine MRI, which will allow the diagnosis of a spinal tumor. Although some features have been described to identify spinal SFT, they are not diagnosed correctly on MRI, because of their rarity, lack of specific features, and variety of radiological presentations [21]. Depending on the radiological type of the lesion, they are misdiagnosed as schwannomas, meningiomas, hemangioblastomas, metastases from solid cancers, ependymomas, and osteosarcomas [3,6,8,13,18,19,20]. Even in cases when a biopsy was performed before surgical resection, in the literature review, the diagnosis was not obtained correctly [18,19]. MRI-based classification [3] has significant limitations in terms of surgical planning, since spinal cord invasion cannot be assessed reliably. As a result, some authors have proposed to classify spinal SFT as vertebral, paravertebral, spinal cord, or mixed, to allow better anatomical understanding [6].

Nevertheless, whether SFT is suspected or not, surgery remains the first-line treatment option (Figure 3). A preoperative CT scan is recommended to assess bone invasion in all cases of spinal tumor diagnosis. The individual decision for arthrodesis is based on tumoral and surgical criteria, including articular process damage or destabilization due to an exceptionally large posterior laminectomy. Spinal angiography is useful for foraminal/anterior tumors located between T8 and L1 to identify the artery of Adamkiewicz, whose lesion can cause definitive paraplegia due to the interruption of the anterior spinal blood supply [22]. Preoperative embolization should be discussed every time spinal SFT is suspected, especially for large tumors or when percutaneous embolization is feasible and has proven to be useful in selected cases [23,24]. Preoperative neurological electrical assessment will rarely have an impact on the surgical decision making, except in pauci-symptomatic patients, balancing in favor of surgery when a neurological impact arises. Perioperative neuro-monitoring is variably available in different hospitals but could be considered whenever intramedullary invasion is suspected. In addition, perioperative ultrasound may be useful in specific cases, when medullar invasion is suspected or to achieve recurrence removal, this technique usually confirming the surgeon’s own microsurgical observation. There is evidence that 5-amino-levulinic acid induces fluorescence in spinal SFT, as in several other tumor types, which could help to identify the limits of invasive tumors, but its clinical usefulness needs to be proven [25]. In our experience, perioperative frozen histological analysis is seldom conclusive, but may rule out other diagnoses. As for any spinal tumor, the goals of surgery are to decompress the neurological structures and safely achieve resection, if possible complete [26]. The patient should be informed of the possible subtotal surgery and need for second-step surgery or adjuvant treatment.

### 4.2. Postoperative Decision Making in Spinal SFT Treatment

Two main questions will arise at this point: First, what information can be reliably given to the patient concerning the tumor recurrence? Second, should any additional treatment be performed after surgery? Spinal SFT are extremely rare tumors. However, they are part of the SFT spectrum, which includes more common locations, including the rare intracranial SFT, and the more frequent pleural SFT [2,27]. All these tumors share the same genomic [2] and transcriptomic [28] identity, although they develop in different organs. Discussing these cases in multidisciplinary meetings with oncologists and surgeons aware of their histological rather than anatomic specificities can be of great help.

One factor that the patient must be aware of is the need for long-term follow-up, with recurrences happening up to 25 years after the initial surgery. Recurrences in meningeal SFT will happen in at least 37% of cases, after a mean 4.7 years, and symptomatic metastases in 10% of cases [26]. In spinal SFT, recurrences happen in 43% of patients after a mean 5.8 years and metastases in 11–25% [3,6]. Radiological or clinical follow-up is usually performed from every 3–6 months in the first few years after surgery to every 5 years life-long in the absence of any event. There is no indication to screen for asymptomatic metastases. However, it may be relevant to keep in mind that other meningeal localizations may occur, since up to two thirds of metastases actually are secondary intracranial or spinal SFT [3,12], and, in our experience, carcinomatous meningitis can also develop. Overall, the 5-year survival rate ranges between 76% and 93% [3].

Prognostic risk factors for recurrence that could help to decide about adjuvant treatment are controversial throughout the SFT literature. Recurrences in SFT in general are more likely to happen in tumors with a high diameter (superior to 6 cm), histological grade, necrosis, high mitotic rate, subtotal resection, absence of postoperative radiotherapy, and some localizations, including central nervous system [27,29,30]. From a molecular point of view, some types of NAB2–STAT6 fusions be associated with a worse prognosis, though not systematically [31]. Screening for the fusion type is not performed routinely and these results are not significant enough to make it necessary in clinical practice. Reviews that focus on spinal SFT, including us, failed to confirm any of these prognostic factors [3], except for subtotal resection in one study, which was significantly associated with a shorter recurrence-free survival and overall survival [6]. This review also identified WHO grades II-III as a risk factor for earlier recurrence but not survival, which could be associated with the progression of residual grade I SFT towards grade III [26].

Whether and when to perform adjuvant radiotherapy is still a matter of debate. There is evidence that adjuvant radiotherapy for both extrameningeal and meningeal SFT may improve local control [27,32,33,34]. However, adjuvant radiotherapy does not prevent the development of neuroaxis or peripheral metastases [34]. Compared to extrameningeal SFT, intradural lesions more often lead to subtotal resection, which is the main risk factor for local recurrence. Therefore, adjuvant radiotherapy could be offered after subtotal resection to delay local recurrence, keeping in mind that no effect on survival has been proven [3,6,34]. Moreover, radiation myelopathy, although rare, could significantly alter the quality of life of patients with a long survival. Stereotactic radiosurgery has been used for intracranial SFT, but its use in spinal SFT is sporadic and no conclusion can be drawn. Conventional chemotherapy gives a poor clinical benefit, and anti-angiogenic treatments are the most promising option [35,36,37], used in one patient in our series, as a third-line option.

### 4.3. Specificities of Spinal SFT

Although there is no controversy about the common molecular identity of SFT in all localizations since the description of NAB2–STAT6 fusion [2,28], spinal SFT is an ambiguous concept based on anatomy. They are usually considered meningeal SFT because they cause spinal cord compression and therefore neurological deficits. However, as illustrated by the wide variety of radiological and operative findings, it is not clear where these tumors arise from. Indeed, spinal SFT may well arise from the intradural space, from the vertebra [19,20], or from the pleura. Perioperative findings support the fact that these fibroblastic tumors arise from different layers, with some tumors clearly extramedullary [11], even extradural, whereas others present obvious signs of pial, nerve roots, or even spinal cord invasion [6,12,15]. Whether this variability is a sign of tumor aggressiveness or site of origin is not clear and no histological description of medullary invasion has been reported until now, whereas brain invasion has been reported in intracranial SFT, as in meningiomas [1].

This anatomical ambiguity correlates with the fact that the cell of origin of SFT is not determined: although it was advocated that meningeal SFT arise from a specific prostaglandin-D2-synthase-positive cell type, as with meningiomas [38,39], there is also molecular evidence that meningeal SFT probably share the same mesenchymal origin as all SFT [28]. This encourages us to treat spinal SFT as nonspecific to the central nervous system, questioning the fact that current clinical trials for SFT exclude patients with meningeal tumors.

## 5. Conclusions

Spinal SFT are extremely rare and versatile fibroblastic neoplasms with a high propensity to recur, representing a diagnostic and therapeutic challenge. Since clinical and radiological presentation does not usually allow preoperative diagnosis, surgery remains essential to achieve both diagnosis and neurological decompression. As spinal SFT are unequivocally part of the SFT spectrum in terms of molecular identity, they should be treated as such by a multidisciplinary team rather neurosurgeons alone. However, spinal SFT present specificities owing to its close relationship with the spinal cord. In particular, it seems that radiotherapy should be considered whenever gross total resection cannot be achieved due to spinal cord pial invasion given the significant rate of recurrence. Future developments of targeted therapies or neurologically sparing radiation protocols may help to control these tumors without damaging the surrounding neurological structures.

## Figures and Tables

**Figure 1 cancers-14-02839-f001:**
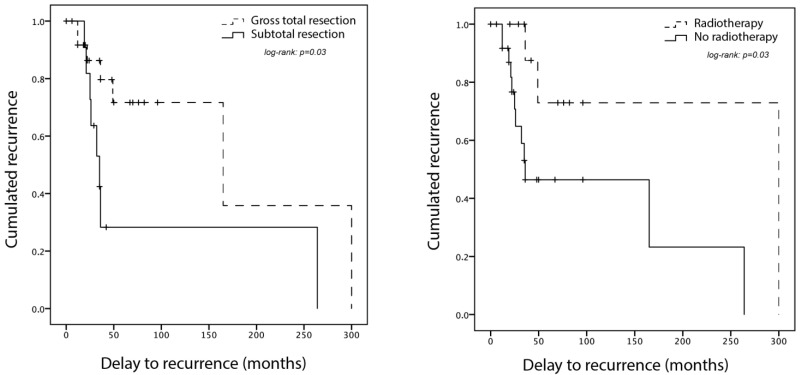
Recurrence-free survival (Kaplan–Meier curve) for all patients with a spinal solitary fibrous tumor and at least 12 months of follow-up, based on our series and literature systematic review (*n* = 35). Each + accounts for a patient death or end of follow-up. Left: for patients with gross total resection compared to subtotal resection. Right: for patients with adjuvant radiotherapy compared to no radiotherapy. Radiotherapy significantly improved recurrence-free survival in patients with subtotal resection.

**Figure 2 cancers-14-02839-f002:**
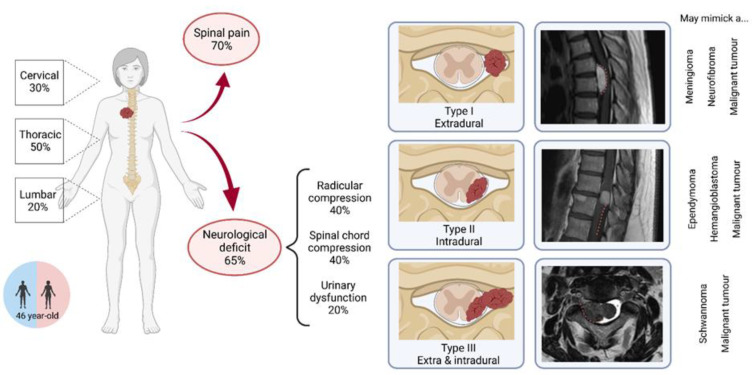
Graphical summary of the characteristics and clinico-radiological presentation of patients with spinal solitary fibrous tumors, including radiological types of SFT on MRI.

**Figure 3 cancers-14-02839-f003:**
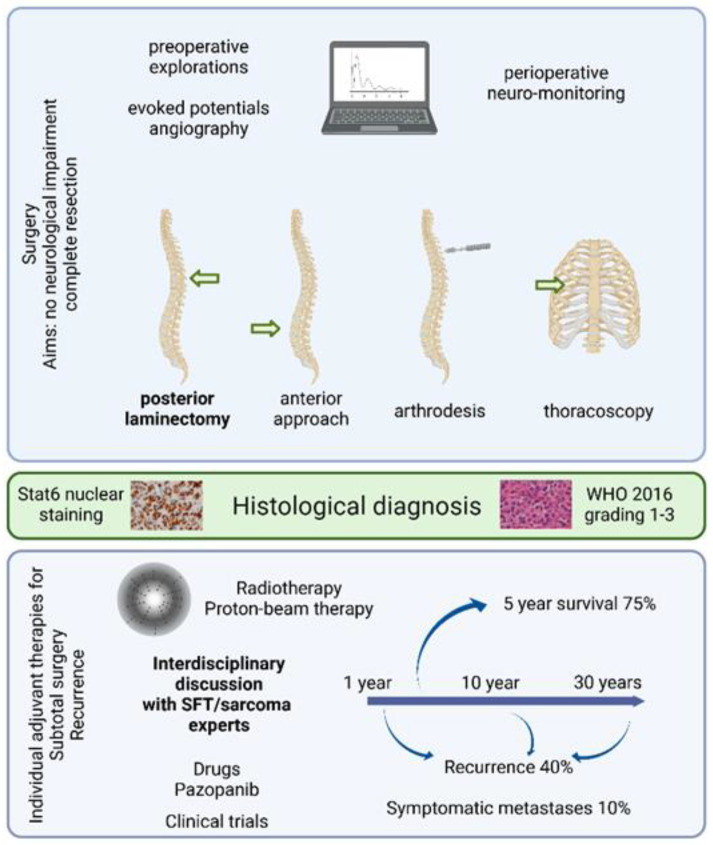
Graphical summary of the surgery, diagnosis, adjuvant treatments, and outcome of patients with spinal solitary fibrous tumors.

**Table 1 cancers-14-02839-t001:** Description of characteristics and prognosis, for the population presenting a spinal solitary fibrous tumor, as diagnosed with positive STAT6 nuclear staining, for our series (*n* = 10) and a systematic review of the literature (*n* = 31).

Characteristics		Our Series (*n* = 10)	Literature Review (*n* = 31)	Total (*n* = 41)
Population				
Sex	M	6 (60%)	14 (45%)	**20 (49%)**
	F	4 (40%)	17 (55%)	**21 (51%)**
Age, mean ± IC95		47 ± 8	45 ± 3	**46 ± 4**
Clinical and radiological presentations				
Spinal pain		5 (50%)	22 (76%)	**27 (69%)**
Radicular compression		6 (60%)	8 (28%)	**14 (36%)**
Spinal cord compression		7 (70%)	9 (31%)	**16 (41%)**
Urinary dysfunction		4(40%)	3 (10%)	**7 (18%)**
Motor dysfunction		5 (50%)	9 (29%)	**14 (34%)**
Sensory dysfunction		4 (40%)	6 (19%)	**10 (24%)**
Duration of symptoms (mo)		10 ± 6	20 ± 12	**17 ± 9**
Tumor localization	Cervical	5 (50%)	7 (23%)	**12 (29%)**
	Thoracic	4 (40%)	17 (54%)	**21 (51%)**
	Lumbar	1 (10%)	7 (23%)	**8 (20%)**
Tumor type	I extradural	0 (0%)	2 (14%)	**2 (9%)**
	II intradural	7 (78%)	8 (57%)	**15 (65%)**
	III extra- and intradural	2 (22%)	4 (29%)	**6 (26%)**
Surgical and histological findings				
Complete resection		7 (70%)	22 (71%)	**29 (71%)**
Purely extramedullary tumor during surgery		5 (50%)	19 (68%)	**24 (63%)**
Histological grading	1	1 (11%)	6 (55%)	**7 (35%)**
	2	5 (56%)	0 (0%)	**5 (25%)**
	3	3 (33%)	5 (45%)	**8 (40%)**
Post-operative management and outcome				
Primary adjuvant treatment	None	6 (60%)	14 (77%)	**30 (73%)**
	Radiotherapy	4 (40%)	7 (23%)	**11 (27%)**
Documented recurrence		4 (40%)	11 (32%)	**15 (37%)**
Time to first recurrence (mo)		128 ± 116	49 ± 42	**70 ± 47**

**Table 2 cancers-14-02839-t002:** Risk factors for recurrence in patients with a minimum 12 months of follow-up (*n* = 35), *p*-value for exact Fisher test. WHO: World Health Organization.

Risk Factor for Recurrence	Recurrence (*n* = 15)	No Recurrence (*n* = 20)	*p*-Value
Intramedullary component	36% (*n* = 14)	26% (*n* = 29)	0.70
Subtotal resection	53% (*n* = 15)	20% (*n* = 20)	0.07
WHO grade 3	40% (*n* = 5)	50% (*n* = 10)	1
Adjuvant radiotherapy	13% (*n* = 15)	40% (*n* = 20)	0.13

## Data Availability

Data supporting reported results can be found in Appendix A.

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
