# Peer review of "Spinal Solitary Fibrous Tumors: An Original Multicenter Series and Systematic Review of Presentation, Management, and Prognosis"

_cancers, 2022, doi:10.3390/cancers14122839_

Round 1

Reviewer 1 Report

Dear Authors

I have reviewed your paper with great interest.

I will accept your paper after a minimal revision.

My revision is:

Title: Very Good

Abstract: Very Good

Introduction and AIM: The problem and the aim are well descripting.

Marterials, Patients and methods and statistics: All good.

Results: Focus on and well described.

Discussion and Thread: effectiveness Focus ON.

Can short spinal stabilization help in these pathologies? cite and discuss the following paper:

Medici A, Meccariello L, Falzarano G. Non-operative vs. percutaneous stabilization in Magerl's A1 or A2 thoracolumbar spine fracture in adults: is it really advantageous for a good alignment of the spine? Preliminary data from a prospective study. Eur Spine J. 2014 Oct;23 Suppl 6:677-83. doi: 10.1007/s00586-014-3557-7. Epub 2014 Sep 12. PMID: 25212447.

Muratori F, Esposito M, Rosa F, Liuzza F, Magarelli N, Rossi B, Folath HM, Pacelli F, Maccauro G. Elastofibroma dorsi: 8 case reports and a literature review. J Orthop Traumatol. 2008 Mar;9(1):33-7. doi: 10.1007/s10195-008-0102-7. Epub 2008 Mar 13. PMID: 19384479; PMCID: PMC2656974.

Righi A, Pacheco M, Pipola V, Gambarotti M, Benini S, Sbaraglia M, Frisoni T, Boriani S, Dei Tos AP, Gasbarrini A. Primary sclerosing epithelioid fibrosarcoma of the spine: a single-institution experience. Histopathology. 2021 Jun;78(7):976-986. doi: 10.1111/his.14332. Epub 2021 Apr 14. PMID: 33428796.

Boriani S, Pipola V, Cecchinato R, Ghermandi R, Tedesco G, Fiore MR, Dionisi F, Gasbarrini A. Composite PEEK/carbon fiber rods in the treatment for bone tumors of the cervical spine: a case series. Eur Spine J. 2020 Dec;29(12):3229-3236. doi: 10.1007/s00586-020-06534-0. Epub 2020 Jul 20. PMID: 32691220.

Colangeli S, Capanna R, Bandiera S, Ghermandi R, Girolami M, Parchi PD, Pipola V, Sacchetti F, Gasbarrini A. Is minimally-invasive spinal surgery a reliable treatment option in symptomatic spinal metastasis? Eur Rev Med Pharmacol Sci. 2020 Jun;24(12):6526-6532. doi: 10.26355/eurrev_202006_21636. PMID: 32633339.

Can scoliosis be induced when ribs are removed? cite and discuss the following paper:

Cervera-Irimia J, González-Miranda Á, Riquelme-García Ó, Burgos-Flores J, Barrios-Pitarque C, García-Barreno P, García-Martín A, Hevia-Sierra E, Rollo G, Meccariello L, Caruso L, Bisaccia M. Scoliosis induced by costotransversectomy in minipigs model. Med Glas (Zenica). 2019 Aug 1;16(2). doi: 10.17392/1015-19. Epub ahead of print. PMID: 31223011.

References: Well chosen but to improve

Figures and Table: Very Good.

Author Response

Dear reviewer,

Thank you for your positive comments and suggestions. To answer your questions, we modified the text as followed. Can short spinal stabilization help in these pathologies? Can scoliosis be induced when ribs are removed?

Spinal destabilization may indeed be an issue in spinal SFT surgery like in any other type of spinal tumor. Cases that require stabilization include articular invasion by the tumor (paravertebral SFT for instance), or articular disruption by the surgeon. In addition, tumors involving several vertebral levels (around four or more) may require extensive laminectomy and stabilization. However, those cases rarely happen for SFT, and there is no case in our series that falls into those categories. In practice, each case must be tackled individually based on preoperative bone CT-scan and surgical approach, just like any other spinal tumor.

Ribs exceptionally need to be removed for SFT surgery, and in those cases the rationale would be the same as for any other spinal tumor. Thoracoscopy does not induce spinal destabilization or scoliosis. Costotomy does not induce scoliosis per se, but corporectomy may cause it, if it involves more than a third of the vertebra. In those cases, especially in transitional segments, intercorporeal arthrodesis using for instance rib graft can be necessary.

To include those comments, we modified the manuscript as follows: “Preoperative CT-scan is recommended to assess bone invasion in all cases of spinal tu-mor diagnosis. Individual decision for arthrodesis is based on tumoral and surgical cri-teria, including articular process damage or destabilization due to an exceptionally large posterior laminectomy.” (l.196). We did not modify the figures, since the rare necessity of arthrodesis is already illustrated in Fig3.

We thank you for helping us improve our article and we hope you will judge it is now suitable for publication. Best regards, Caroline Apra, on behalf of all authors.

Reviewer 2 Report

The authors reviewed a monocentric series of 10 Stat6-positive SFT. They also performed a review of 31 cases retrieved from a systematic literature review. All STF were graded radiological as extradural, intradural, and intra- to extradural and paravertebral type. All tumors were histologically graded according to 2016 WHO classification.

The manuscript is a relevant addition to the current literature, but I have some issues:

which role for advanced intraoperative imaging? For example, intraoperative ultrasound could help to increase the EOR or reduce the risk of recurrence (see i.e. the role of contrast IOUS in intradural and intramedullary tumors,  Vetrano IG et al. Contrast-Enhanced Ultrasound Assisted Surgery of Intramedullary Spinal Cord Tumors: Analysis of Technical Benefits and Intra-operative Microbubble Distribution Characteristics. Ultrasound Med Biol. 2021 Mar;47(3):398-407. doi: 10.1016/j.ultrasmedbio.2020.10.017)

There is a role also for florescent dyes as 5-ALA or sodium fluorescein?

Whereas the whole series has been reviewed to adequate the past classification to the WHO 2016 (when the term SFT/hemangiopericitoma were unified), the 2021 classification has deleted hemangiopericitoma as a nosological entity. Authors could clarify this aspect in the introduction, and remove the term ins the headings and sections, as the tumor is now termed only SFT ( see Louis DN et al. The 2021 WHO Classification of Tumors of the Central Nervous System: a summary. Neuro Oncol. 2021 Aug 2;23(8):1231-1251. doi: 10.1093/neuonc/noab106)

I think that some intraoperative videos and pics, if available, could increase the educational role of the manuscript

Author Response

Reviewer 2

Dear reviewer, thank you for your positive comments and thorough review. Here are our answers. We modified the text as suggested:

  • which role for advanced intraoperative imaging?

Indeed, perioperative imaging, i.e. US (perioperative MRI is not an option currently for spinal tumors), is relevant is some cases. The efficacy of using US still needs to be proven in terms of EOR, as discussed in the reference you suggested, but it remains a helpful tool for surgeons. In our experience, it can be used to orientate the dura opening, and, more interestingly, to check the aspect of the medulla after resection. Its use highly depends on the quality of the US, and on the surgeon’s experience, and cannot be considered a gold standard for all spinal tumor surgery. In addition, it will be difficult to prove its usefulness in SFT, that are an heterogeneous group of tumors, with varied vasculature. Globally, it is a very useful in selected cases, and we reckon it should be mentioned in the review. We modified the manuscript as follows: « In addition, perioperative ultrasound may be useful in specific cases, when medullar invasion is suspected or to achieve recurrence removal, this technique usually confirming the surgeon’s own microsurgical observation.” (l.205)

  • There is a role also for florescent dyes as 5-ALA or sodium fluorescein?

Thank you for asking. 5-ALA and fluorescein are used as fluorescent perioperative markers for glioblastoma resection, and their use in other tumors is still not validated in clinical practice, although they have been studied in all tumor types, especially in metastases. Indeed, 5-ALA has been shown to induce fluorescence in spinal hemangiopericytomas (Analysis of 5-aminolevulinic acid-induced fluorescence in 55 different spinal tumors, Millesi) and may therefore be a useful tool for specific cases. Whether it will give a benefit to the patients needs to be explored. In non invasive SFT, this will not add information, but in invasive cases, it may help surgeons understand the localization of the tumor. As for any other spinal tumor, there will be a surgical difficulty to choose between resection of invasive fluorescent tissue and protection of the medulla. In priority, the aim of surgery is to avoid any neurological worsening, whereas complete resection is a secondary objective. We modified the manuscript and added a reference (Millesi): “There is evidence that 5-amino-levulinic acid induces fluorescence in spinal SFT, like in several other tumor types, which could help identify the limits of invasive tumors, but clinical usefulness needs to be proved [25]”(l.208).

  • Whereas the whole series has been reviewed to adequate the past classification to the WHO 2016 (when the term SFT/hemangiopericitoma were unified), the 2021 classification has deleted hemangiopericitoma as a nosological entity.

Thank you for underlying this update, we modified the manuscript and the title to remove “HPC” and modified the introduction as follows: “In 2016, the World Health Organization (WHO) introduced the combined term “solitary fibrous tumor / hemangiopericytoma” for describing connective tissue tumors of the central nervous system with positive Stat6 nuclear immunostaining, which was replaced by “solitary fibrous tumors” (SFT) alone in 2021, to conform fully with soft tissue pathology nomenclature[1].” We added the 2021 reference (Louie).

  • I think that some intraoperative videos and pics, if available, could increase the educational role of the manuscript

Thank you for suggesting. We added a supplementary perioperative video under microscope. “A video showing perioperative observations is available as a supplementary document ”. (Results).

We thank you for helping us improve our article and we hope you will judge it is now suitable for publication. Best regards, Caroline Apra, on behalf of all authors.